# New Insights into Interspecific Hybridization in *Lemna* L. Sect. *Lemna* (Lemnaceae Martinov)

**DOI:** 10.3390/plants10122767

**Published:** 2021-12-15

**Authors:** Luca Braglia, Diego Breviario, Silvia Gianì, Floriana Gavazzi, Jacopo De Gregori, Laura Morello

**Affiliations:** Istituto Biologia e Biotecnologia Agraria, Via Alfonso Corti 12, 20131 Milano, Italy; braglia@ibba.cnr.it (L.B.); breviario@ibba.cnr.it (D.B.); giani@ibba.cnr.it (S.G.); gavazzi@ibba.cnr.it (F.G.); degregori@ibba.cnr.it (J.D.G.)

**Keywords:** duckweeds, Lemnaceae, interspecific hybrids, tubulin-based polymorphism, β-tubulin

## Abstract

Duckweeds have been increasingly studied in recent years, both as model plants and in view of their potential applications as a new crop in a circular bioeconomy perspective. In order to select species and clones with the desired attributes, the correct identification of the species is fundamental. Molecular methods have recently provided a more solid base for taxonomy and yielded a consensus phylogenetic tree, although some points remain to be elucidated. The duckweed genus *Lemna* L. comprises twelve species, grouped in four sections, which include very similar sister species. The least taxonomically resolved is sect. *Lemna*, presenting difficulties in species delimitation using morphological and even barcoding molecular markers. Ambiguous species boundaries between *Lemna minor* L. and *Lemna japonica* Landolt have been clarified by Tubulin Based Polymorphism (TBP), with the discovery of interspecific hybrids. In the present work, we extended TBP profiling to a larger number of clones in sect. *Lemna*, previously classified using only morphological features, in order to test that classification, and to investigate the possible existence of other hybrids in this section. The analysis revealed several misidentifications of clones, in particular among the species *L. minor*, *L. japonica* and *Lemna gibba* L., and identified six putative ‘*L. gibba*’ clones as interspecific hybrids between *L. minor* and *L. gibba*.

## 1. Introduction

The genus *Lemna* L. (Lemnaceae, Martinov) [1] is thought to have originated around 41.7 MYA (crown age) from a common ancestor which separated from the Wolffioideae Engl. (genera *Wolffia* Horkel ex Schleiden and *Wolffiella* Hegelm.) branch around 54.4 MYA [2]. Its most probable origin has been established as around 16.4–41.7 MYA in North America. According to the latest taxonomic revision, uniting *Lemna minuta* Kuntz with *Lemna valdiviana* Phil., the genus *Lemna* comprises 12 species [3], grouped in four monophyletic sections: *Alatae* Hegelm., *Uninerves* Hegelm., *Biformes* Landolt, and *Lemna* [4]: this categorisation into sections is also supported by GBS data [5]. Sect. *Lemna* includes seven species, among which we can find the most cosmopolitan *Lemna gibba* L. and *Lemna minor* L., as well as geographically restricted species such as *Lemna disperma* Hegelm. (Oceania) and *Lemna obscura* (Austin) Daubs (South-East coast of North America and Ecuador). This is the most problematic section within the taxonomically complex genus *Lemna*, in which boundaries between some species seem to blur, due to extremely similar morphology. Species can be distinguished from each other based on few, recently updated, key features [3], some of which refer to flowers or fruits, rarely observed in many species. Although sufficient in most cases, key features may vary within the same species among clones or under different growth conditions, particularly upon in vitro cultivation, making identification difficult.

For example, *L. gibba* is usually easily recognized for the inflated (gibbous) form of the frond caused by enlarged air spaces in the aerenchyma tissue. However, morphological variability in *L. minor* and *L. gibba* has long been known, and discrimination between the two becomes particularly difficult when air spaces of *L. gibba* are reduced, making fronds flat [6,7,8]. This has led to grouping the two species in the *L. minor*–*L. gibba* group, or complex, which identifies a continuum between the two [8,9]. Both species are long-day plants which sometimes share the same habitat. Variations in frond morphology in *L. gibba* may be seasonal. but the occurrence of mixed populations of gibbous and permanently ‘flat forms’ of *L. gibba* (*L. minor*-like) has been repeatedly reported in The Netherlands, together with supposed transition forms between the two species, which could not be assigned with certainty to either of the two [9,10].

Species with intermediate traits between *L. minor* and *L. gibba*, but bearing some distinctive features have been described in the past, such as *Lemna parodiana* Giardelli in Argentina [11] and *Lemna symmeter* Giuga, in Southern Italy [12]. In the absence of further evidence, such species have been considered conspecific with *L. gibba* [13].

Intraspecific variability in *L. minor* is represented by reported differences in chromosome numbers, from 40 to 50, and by different genome size reported among clones [14,15], although data are not always consistent because of the small chromosome size and variability in genome size estimation by different methods of measurements. It is therefore difficult to say how much the wide morphological intraspecific diversity observed could be due to phenotypic plasticity or to genetic diversity.

*Lemna disperma* shows a combination of characters of *L. gibba* and *L. minor* [16], but it is restricted to Australia and New Zealand. Similarly, *L. obscura*, was previously identified either as *L. minor* or *L. gibba* but is limited to temperate regions of North America [16]. In both cases, morphological classification is often supported by geographical distribution. A further species in this section, former *Lemna ecuadoriensis* Landolt, is now considered conspecific with *L. obscura* [17].

Moreover, *Lemna turionifera* Landolt, of Northern Asia and America, can be occasionally confused with *L. minor* and *L. gibba* by frond morphology, but fruits and seed characters, together with turion-forming ability, provide distinctive traits. *Lemna japonica* was described as a new species in 1980 as a biotype of *L. minor* with a limited geographical distribution and posited as a possible interspecific hybrid between *L. minor* and *L. turionifera* [18]. This hypothesis was supported by intermediate allozyme pattern shown by *L. japonica* clones with respect to other similar specimens collected in Japan and likely corresponding to *L. minor* and *L. turionifera* respectively [19]. This evidence was recently supported by genetic proofs based on intron length polymorphism in the β-tubulin genes (also known as Tubulin Based Polymorphism, TBP) and AFLP. Accordingly, *L. japonica* is hereafter indicated as *Lemna* ×*japonica* to indicate the hybrid status of this taxon, assessed as *L. minor*-SubCluster II (*L. minor sensu lato*) in our previous work [20]. Its similarity with *L. minor* is therefore evident and is the cause of frequent misclassifications.

The species most recently included in the section is *Lemna trisulca*, formerly separated in the single-species sect. *Hydrophylla* Dumort. This species has a unique morphology with submerged, narrowly ovate fronds connected to a green stalk, often forming branched chains. Despite this peculiarity, the combined data cladogram (morphological, flavonoid, allozyme and DNA sequence data) first obtained by Les [4] clearly placed *L. trisulca* within sect. *Lemna*, as later confirmed by nuclear and plastid molecular markers [21]. In some cases, even the more distantly related species like *L. minuta*, native to America but invasive in Europe [22], can be distinguished from the European native *L. minor* only by quantitative morphometry [23]. The advent of molecular taxonomy has greatly facilitated species delimitation among duckweeds by AFLP fingerprinting [24,25] and plastid barcoding sequences as *psbK-psbI* and *atpF-atpH*, which are now commonly used for accurate identification of clones [26,27]. Nuclear and plastid molecular markers have also bolstered phenetics in improving phylogenetic studies [4,21] on duckweeds. However, despite this progress, phylogenetic uncertainty still persists among some lineages and some nodes resolved incongruently by using plastid and nuclear ribosomal sequences [21]. This has been repeatedly attributed to potential interbreeding or incomplete divergence, although neither has ever been demonstrated. The application of high-throughput methods as genotyping-by-sequencing (GBS) has helped to resolve those problematic species boundaries in the genus where plastid sequences alone were inadequate as in the case of *L. minor* and *L.* ×*japonica* [5]. This observation is in accordance with the maternal inheritance of plastids from *L. minor*, in the light of the finding that *L.* ×*japonica* is of hybrid origin [20]. On the one hand, some interpretive problems may have arisen because some accessions were not identified accurately [4] while, on the other, when many clones of the same species were compared, molecular marker analysis enabled uncovering possible misclassifications of clones [20,24,26], particularly in sect. *Lemna*.

To this end, TBP fingerprinting has been particularly suitable for species delimitation in the genus [20]. This genetic profiling method had been successfully applied to species and subspecies level discrimination in different plant families [28,29,30]. The Landolt Duckweed Collection (LDC, Rotterdam, The Netherlands, http://www.duckweed.ch (accessed on 8 June 2021)) is perhaps the most important historical collection worldwide, with over 500 clones left of the more than 1000 collected and morphologically classified over 70 years by Professor Elias Landolt in Zürich, CH. Many replicated clones are also present in the collections of the Rutgers Duckweed Stock Cooperative (http://www.ruduckweed.org/ (accessed on 14 December 2021)) and at the University of Jena and are being investigated in many laboratories worldwide. Although many clones have been investigated by molecular markers, a large part of the collection remains genetically unexplored. As TBP profiling provides a simple way for duckweed species discrimination without sequencing, we planned to investigate all LDC’s clones, under an agreement with Mr. W. Lämmler, the manager of the LDC.

We started with a large selection of clones, about 100 in the problematic *Lemna* sect. *Lemna*, with the dual aim of verifying the morphological classification of each clone and of finding evidence for the existence of other interspecific hybrids in this section. Given the importance of the LDC as a fundamental resource for scientists working in the field, our data provide useful information for further investigations and for a critical revision of the literature. Interspecific relationships within sect. *Lemna* are also investigated by length and sequence similarity in β-tubulin introns. Moreover, leveraging the high genetic variability of such regions, introns are also used as a suitable source of SNPs for the evaluation of intraspecific variability.

## 2. Results

### 2.1. TBP Profiling of Duckweed Clones in Lemna Sect. Lemna

TBP was demonstrated to be a reliable tool for clustering *Lemna* clones according to the respective species, as validated by plastid markers [20]. Distinctive amplification profiles are obtained for each species, with some intraspecific allelic variations. Ninety-eight duckweed clones belonging to sect. *Lemna* were analyzed by TBP profiling of the first and second β-tubulin introns (Table 1 and Appendix A). Fifty-seven clones belonging to the same section and analyzed in the previously mentioned work [20] were added to the cluster analysis, with *Landoltia punctata* clone 9354 used as an outgroup. The scoring of the Capillary Electrophoresis TBP (CE-TBP) peaks revealed 139 polymorphic markers across the seven *Lemna* species (87 and 52 from the 1st and the 2nd intron region, respectively). The derived dendrogram is shown in Figure 1.

The separation of the seven *Lemna* species forming the section, according to their genetic similarity by TBP, allowed the unequivocal reclassification of those clones which do not correspond to their morphological description. The hybrid status of *L.* ×*japonica* is confirmed here by this larger dataset. In fact, all the clones in this cluster, which includes the *L.* ×*japonica* holotype 7182, showed hybrid TBP profiles between *L. minor* and *L. turionifera*, reflecting the duplicate set of six β-tubulin genes, whereas all clones in the *L. minor* cluster have similar pattern among each other, with just six main peaks. Despite the low bootstrap values (<50%) of this branch, the tree topology is clearly due to the fact that *L.* ×*japonica* shares alleles with both putative parental species *L. minor* and *L. turionifera*. Interestingly, through the whole tree, high probability support was given to some sub-clusters indicating intraspecific allelic variance among populations (in Figure 1). Separation was in agreement with the geographical origin of clones, as in the case of the American clusters of *L.* ×*japonica*, *L. turionifera* (the only American clone) and *L. gibba,* the East Asian clusters of *L.* ×*japonica* and *L. turionifera* and a Mediterranean group of *L. gibba* (Figure 1).

The TBP data were also used to infer a principal component analysis (PCA) to further describe the genetic diversity among clones belonging to the four most represented species, *L. minor, L.* ×*japonica, L. gibba* and *L. turionifera* (Figure 2). The cumulative contribution of the first three principal components explains 63% of the total variation, providing a clear description of the relationships among species, concurrently revealing hybrid entries, which grouped separately. This was the case of *L.* ×*japonica* clones, including those originally classified as *L. minor*, which were clearly separated from the putative parental species *L. minor* and *L. turionifera,* along the plot axes.

Similarly, an additional group, with respect to the four recognized species, was formed by six clones, mentioned above as the Mediterranean group, classified as *L. gibba* by morphology and representing one of two sister clades of the *L. gibba* cluster observed in the dendrogram of Figure 1. The bi-dimensional plot placed this group of clones in an intermediate position between *L. minor* and *L. gibba*, suggesting shared alleles with both species. This prompted us to further investigate if the aforementioned group of six clones could be considered a separate taxon, possibly a hybrid. In the plot and thereafter we then refer to this group as *Lemna gibba* × *Lemna minor* (see below).

In addition, within each species, isolated subgroups can be recognized as spread apart from the main clusters by the first three components of the PCA (Figure 2). In accordance with the dendrogram in Figure 1, the subgrouping distribution shown by the PCA is congruent with the geographical origin of clones (colored dots in Figure 2). Notably, in *L. minor* a group of clones from the Middle East and Africa was significantly spread apart from the respective main group.

### 2.2. Reclassification of Clones by TBP

The correspondence between the original morphological characterization of the analyzed clones and TBP results is summarized in Table 2, which reports misidentifications for each species and the kind of error involved. *L. obscura*, *L. trisulca* and *L. disperma* were the most easily identified species with a 100% correct assignment, although a reduced number of clones was available, and results are not included in the Table. The overall misidentification rate by morphology was 28.6% considering five species (treating *L. gibba* × *L. minor* hybrid as a separate taxon). Classification by morphology failed to correctly identify 48% of *L.* ×*japonica* clones, considering them as *L. minor* (30%), *L. gibba* (14%), and *L. turionifera* (5%). Fifty-five percent of supposed ‘*L. gibba*’ turned out to be either *L.* ×*japonica* (25%) or the newly described hybrid *L. minor × L. gibba* (30%). In just two cases each, clones of *L. turionifera* and *L. gibba* were exchanged for *L.* ×*japonica*. It is therefore clear that most incorrect identifications involved hybrids, which were often classified as one of the parental species. The new classification of all clones according to TBP is given in Table 1.

The high rate of misidentification of *L.* ×*japonica* has at least two direct implications:The abundance of *L.* ×*japonica* populations was highly underestimated, as well as its geographical distribution, which is not limited to Japan, Korea and the east coast of China, as reported by Landolt for *L. japonica* [16]. The actual distribution, deduced from investigated clones and shown in Figure 3, covers all the temperate regions from Eastern Asia to Central Asia, Europe and North America, although their invasive origin in the different regions remains to be elucidated. One clone was even found in South Africa (Figure 3).At least part of the huge variability observed in *L. minor*, e.g., in genome size, ploidy or physiological parameters etc. could be due to erroneous classification of clones.

### 2.3. Lemna gibba × Lemna Minor Hybrids

Comparison of electrophoretic TBP profiles of the six clones in the *L. gibba* subcluster with those of other species revealed additional peaks attributable to *L. minor*, definitively confirming the hypothesis of interspecific hybridization between the two species. As shown in Figure 4, the additivity of the peak profiles in the hybrid clones witness the presence of two subgenomes. However, karyotyping will be needed in order to determine if hybrids are allotetraploids, as usually expected for interspecific hybrids, or homoploids, as it can be the case for asexually reproducing plants. Each TBP amplicon was assigned to the correspondent β-tubulin locus, based on the sequences retrieved from Whole-genome sequencing (WGS) data of *L. minor* 5500 [31] (https://genomevolution.org/r/ik6h (accessed on 14 November 2021), ID 27408), *L. minor* 8627, here reclassified as *L.* ×*japonica,* and *L. gibba* 7742a (https://www.lemna.org/ (accessed on 14 November 2021)). The two sets of six parologous *TUBB* loci in *L. minor* and *L. gibba* were arbitrarily numbered *TUBB*1-*TUBB*6, in the absence of rules for tubulin gene numbering in plants. Corresponding positions on chromosomes or contigs of WGS data are given in Appendix A. An additional β-tubulin sequence, likely a pseudogene, was retrieved only from the *L. gibba* WGS data. In fact, its sequence lacks the canonical two introns and has a short deletion in the second exon, leading to its interpretation as a retrotransposed copy of *TUBB*2 by sequence similarity, and therefore named Ψ*TUBB*2. This sequence is nevertheless amplified by TBP primers, producing a short fragment of 250 base pairs distinctive of *L. gibba* (Figure 4).

The possibility of an artefact originating from the analysis of cross-contaminated clones was excluded by sub-cloning twelve single fronds of each clone and performing the TBP analysis on each clonal population after one week’s cultivation. Profiles were identical to those of the original clones (not shown). On this base, we therefore concluded that we have identified a new interspecific hybrid in the genus, between *L. minor* and *L. gibba*, to add to *L.* ×*japonica*.

Plastid intergenic spacers *psb*K-*psb*I and *atp*F-*atp*H were used to identify the maternal parent of each putative hybrid clone. Interestingly, both reciprocal crosses were observed: clones 9425a and 9248 have maternal inheritance of *L. gibba*, whereas plastid markers of clones 7641, 7320, 6861, 9562 matched the *L. minor* sequences (Appendix A). This seems different from what was found in *L.* ×*japonica*, where all clones so far investigated by plastid barcoding sequencing have *L. minor* as the maternal parent. Interestingly, the two reciprocal crosses were separated by cluster analysis. 

### 2.4. Frond Morphology and Flowering

All six *L. gibba* × *L. minor* clones showed a flat morphology of the frond under standard culture conditions. Frond shape was quite heterogeneous among clones, from almost round to ovate, generally asymmetrical. An overview is given in Figure 5. They were also different in size, with the two *L. minor* × *L. gibba* (9425a and 9248) showing larger fronds than the reverse crosses (Average length 4.45 vs. 3.88 mm¸ average width 3.01 vs. 2.62 mm). We will present elsewhere a more detailed, formal description of the new hybrid species *L. gibba* × *L. minor*.

Descriptions of supposed intermediate forms between *L. gibba* and *L. minor* are reported in the literature. In particular, a new species from Southern Italy was described in 1973 as a possible hybrid between the two species and was named *L. symmeter* Giuga, species nova [12] is now considered as a synonym of *L. gibba* [32]. The name comes from the most distinctive trait reported for the new species, which is the symmetric (simultaneous) development of stamens during flower development. This is different from what described by Kandeler [7] and by Giuga himself about flowering in *L. gibba*, characterized by the appearance of the first anther together or soon after the stigma, later followed by the second anther. The new species was described as sterile, as fruit formation and seed setting was never observed.

We tried to verify if the hybrid clones that we have identified could correspond to putative ‘*L. symmeter*’ on these criteria, by inducing flowering through treatment with salicylic acid (SA) as reported by others [33,34]. After four weeks of cultivation in 20 μM SA, we were able to induce flowering in two out of three *L. gibba* clones (7742a and 9598, not 8124), but neither the six hybrid clones, nor two *L. minor* clones (9977 and 9942) showed reproductive organs after six weeks. Therefore, we have not yet been able to provide proof of the identity of the hybrid clones with the previously described species *L. symmeter.*


### 2.5. Infrasectional Structure of Lemna Sect. Lemna

The infrageneric structure of the genus *Lemna* is not unequivocally defined because of conflicting results obtained by the nuclear ribosomal coding and noncoding sequences, and plastid markers [21] and/or possible misclassification of some clones used for these studies. Nevertheless, a phylogenetic tree was obtained by merging a large number of sequence data [4,21]. The finding of hybrids calls for an update to that tree. Fast evolving intron regions are suitable for phylogenetic analysis of closely related taxa [35], particularly if a low recombination rate is expected, as in the case of predominant clonal propagation in duckweeds. We chose two β-tubulin introns showing little or no intraspecific length variability by TBP profiling and designed gene-specific primers for cross-species amplification, at their exon-intron borders. After optimization of primers and PCR conditions, the first intron of *TUBB*2 and the second intron of *TUBB*1 were amplified in all species of the section with the same two primer pairs. PCR amplification is shown in Figure 6, which also highlights the hybrid origin of *L.* ×*japonica* and *L. gibba* × *L. minor* by the presence of markers of the same length of those shown by the two parental sub-genomes. Intron size was quite conserved among the different species, with the exception of the second *TUBB*1 intron of *L. gibba* and *L. obscura*, showing large deletions. No amplification was obtained on clones of other *Lemna* sections with the same primers (not shown).

Gene trees were obtained by concatenation of the two regions (sequences are provided in Appendix A). Intron sequences of both hybrid species, almost identical (>99% identity) to both parental species, were not included in the alignment. The *TUBB*2 sequence of *Spirodela polyrhiza*, was used to root the tree. Equivalent tree topologies were obtained by both Maximum Parsimony and Maximum Likelihood clustering methods (not shown). The gene tree is congruent with that proposed by Tippery 2015 [21] showing *Lemna* sect. *Lemna* as split in two subclusters, one comprising *L. gibba*, *L. disperma* and *L. minor* and the second *L. turionifera*, *L. trisulca* and *L. obscura*. In that tree, *L.* ×*japonica* was the closest relative of *L. minor*, for the obvious influence of the *L. minor* plastid sequences. A phylogenetic reconstruction of the section based on the *TUBB* gene tree and showing the origin of the two interspecific hybrids is shown in Figure 7.

### 2.6. Intraspecific TUBB2 Polymorphism in Lemna gibba and Lemna minor

As clones from different geographical origin clustered separately by TBP, we searched for intraspecific polymorphisms in *TUBB*2 first intron sequences by investigating 18 clones of *L. minor* and 13 clones of *L. gibba* (Appendix A). PCR amplicons were directly sequenced. All *L. minor* and most *L. gibba* (supposedly diploids), were homozygous at the *TUBB*2 locus, generating unique sequence profiles. Sequences of six *L. gibba* clones could not be obtained due to the superimposition of two separate sequence chromatograms within the same frame, likely arising from length variant heterozygosis. We did not further investigate these clones. Fourteen polymorphic sites out of 410 positions investigated in *TUBB*2 were identified among *L. minor* clones, producing four different sequence variants, here referred to as haplotypes (M1–M4). In *L. gibba*, eight polymorphic sites were found over 380 base pairs, defining three haplotypes, named G1–G3 (Table 3).

The *TUBB*2 sequences of clones *L. minor* 5500 and *L. gibba* 7742a, retrieved from WGS data, were used as references (M1 and G1) for the calculation of the number of polymorphisms. Association of the different haplotypes to each clone is reported in Table 4. In *L. minor*, the separation of an African genetic pool was evident from the sharing of twelve conserved SNPs defining haplotype M2 among the three investigated clones, evident, whereas the Euro-Asiatic clones showed by far larger sequence identity, with 1–2 SNPs.

In *L. gibba,* two clones from the American lineage, genetically distinct from others clones (Figure 1 and Figure 2), showed the same holotype G3, characterized by the presence of six SNPs (Single Nucleotide Polymorphism). 

We also looked for parental genome signatures in the six clones *L. gibba* × *L. minor* by sequence analysis of the *TUBB*2 homoeologous loci, after amplification with selective primers for the two subgenomes G and M. Interestingly, five out of six clones showed the same haplotype at both loci suggesting their common origin and excluding contributions from the American *L. gibba* and African *L. minor* genomic pools. The sixth clone differs for its M haplotype. The high degree of sequence identity between hybrids and the parental species suggests a quite recent origin, with respect to the differentiation of *Lemna* species. 

## 3. Discussion

Species delimitation by morphology can be highly challenging in duckweeds. Within the genus *Lemna* sect. *Lemna* is the most problematic, as it includes closely related species which show blurred boundaries due to high intraspecific and low interspecific variability of morphological traits, as *L. japonica, L. minor and L. gibba*. Even plastid barcoding markers, easily discriminating *L. minor* from *L. gibba*, fails to separate *L. japonica* from *L. minor*. In this work, we extended a previous molecular survey by TBP to 98 additional duckweed clones, mostly from the Landolt Duckweed Collection, classified by morphology as belonging to species of sect. *Lemna*. The analysis provided further support to the identification of *L.* ×*japonica* as a hybrid and led to the identification of a new interspecific hybrid within this section. This is clearly shown in the PCA plot obtained from the TBP dataset, with clear separation of two clusters of hybrid clones sharing *L. minor* as the donor of one subgenome and having *L. turionifera* and *L. gibba*, respectively, as the other parental species.

### 3.1. Lemna ×japonica

Natural interspecific hybrids between *L. minor* and *L. turionifera* were identified in our previous work [20] by TBP cluster analysis as a subcluster merging accessions classified as *L. japonica* and *L. minor* by morphology and nested within the *L. turionifera* branch. This larger analysis, including many more *L. turionifera, L. minor* and *L. japonica* clones, further supported previous evidence that all clones in this cluster are definitely genetically separated from true *L. minor*, despite their similar morphology. All of them have a duplicate set of six β-tubulin genes and highly similar TBP profiles (Appendix A). Therefore, we suggest referring the whole taxonomic unit to *L.* ×*japonica*, to account for its hybrid ancestry. Moreover, TBP profiling is a suitable method to distinguish *L.* ×*japonica* from *L. minor* on a molecular base, in addition to morphology. Accordingly, thirty-six percent of ‘*L. minor*’ clones was identified as *L.* ×*japonica* by TBP analysis. Misidentifications of *L.* ×*japonica* with *L. minor* clones could not be easily recognized before, as plastid markers are not able to resolve the two species [27] and GBS, while detecting some genetic distance between the two species, is not manageable for single clone identification [5].

Although more powerful genetic approaches, including sequencing and karyotype analysis, must be used to address this point, a certain degree of genetic variability within *L.* ×*japonica* is already evident from TBP analysis, separating East Asian and American subclusters (Figure 1) distinct in some private alleles. The species *L. japonica* originally described by Landolt 1980 [18] as the *L. minor* biotype originally present in Japan and Eastern China could possibly coincide with the East-Asian subcluster. Accordingly, all *L.* ×*japonica* clones from China and Japan were correctly classified by morphology, suggesting that either they have more canonical traits or that the geographical origin was a determinant for classification. As a further consequence of misclassification, *L.* ×*japonica* has a broader geographical distribution than previously reported for *L. japonica* by Landolt, as it is present in all temperate regions of Eurasia and North America.

Misidentification of some *L.* ×*japonica* clones as *L. minor* could also partially explain the large variability in genome size (ranging from 356 to 604 Mb) and the variable chromosome numbers, 40–50, attributed to *L. minor* [14,15,36]. For example, at least four *L. minor* clones having higher DNA content according to Wang [14] (550–600 Mb; clone 9016, 9436b, 9439 and 9440) are classified as *L.* ×*japonica* by TBP (the present work), in line with the genome size of the *L. japonica* holotype clone 7182, assessed as 600 Mb [14]. Conversely, other clones also identified as *L.* ×*japonica* by TBP (7210, 7123 and 8434) showed a lower DNA content, around 400 Mbp. However, these clones have different geographical origin, coming either from the American (7123 and 8434) or the African continent (7210).

One possible explanation of these results is that hybridization between *L. minor* and *L. turionifera* has occurred more than once, thus originating independent lineages of *L.* ×*japonica* with different ploidy or chromosome rearrangements as reported for other plant genera, e.g., *Senecio* spp. [37,38], or *Tragopogon* spp. [39].

However, large differences in genome size can also be produced by different proliferation of Transposable Elements (TEs) in different lineages with the same ploidy level. This is widely documented for sunflower (*Helianthus annuus*) where three different homoploid hybrid species, independently originated from the same two parents, featured genome sizes larger then parental species (approximately 5.3–5.6 Gb with respect to 3.3 and 3.5 Gb), mainly represented by TEs [40]. Further work is ongoing in order to understand if *L.* ×*japonica* is monophyletic or may have originated from multiple hybridization events.

### 3.2. Lemna gibba × Lemna minor

The second interspecific cluster that was separated by the PCA plot consisted of six clones, all recorded as *L. gibba* by morphology, with overlapping TBP profiles between *L. minor* and *L. gibba*. While waiting for a formal description including morphological traits (manuscript in preparation), we will refer to it as *L. gibba* × *L. minor* even knowing that the six hybrid clones have originated from both reciprocal crosses of the two parental species.

The finding of this new hybrid was not completely unexpected as intermediate phenotypes between *L. minor* and *L. gibba* were already described in the past [7,8,9,11]. The difficulty in distinguishing these two species, when not flowering, has been widely reported in the literature, particularly when individuals of *L. gibba* do not show the typical gibbosity. In facts, ‘flat forms’ of *L. gibba* often occur under culture conditions or seasonally, at some sites in the wild. Confusion may also arise between *L. gibba* and *L*. ×*japonica*, as shown in our survey. 

We believe that the *L. gibba* × *L. minor* hybrid could correspond to a *Lemna* species described as ‘*L. symmeter*’ [12]. The description of this new species was reported in a monographic publication, written in Italian with only the abstract in English, entitled “Vita segreta di Lemnacee” (The secret life of Lemnaceae) [12]. According to the description, the new *Lemna* species was first observed by the author in 1968, along the coast of Campania (Southern Italy) and later at different sites in Southern Italy where, over the many years, was always found in mixed populations with *L. gibba* (30–90%), rarely associated with *L. minor* or *L. trisulca*. In these mixed populations, *L. gibba* could easily be distinguished by the pronounced gibbosity, typical of most Italian genotypes. Since then, the existence of ‘*L. symmeter*’ has never been confirmed and the original clones went missing. Since the description was not supported by a formal diagnosis in Latin, the putative new species is not valid and ‘*L. symmeter*’ is currently reported as a synonym of *L. gibba*. According to Giuga’s description, flower development becomes the most distinctive trait in the absence of gibbosity. Flowering of ‘*L. symmeter*’ was reported to differ from that of *L. gibba* for the simultaneous growth of the two anthers, occurring a few days after the stigma. Moreover, anthers were reported as indehiscent and seed setting, frequent in *L. gibba*, has never been observed.

Unfortunately, we have so far been unable to induce flowering in any of the hybrid clones to confirm the most important diagnostic trait described by Giuga. However, the present work has demonstrated, by molecular markers and by nuclear sequence analysis that a hybrid species compatible with the description of ‘*L. symmeter’* indeed exists, and is unlikely to be rare in the wild as six clones are present in the Landolt collection. All clones but one came from the Mediterranean basin, three of them (9248, 9562 and 6861) from different Italian regions, from North to South, supporting their correspondence with putative ‘*L. symmeter’*. This is also supported by a recent survey in Central Italy, in which 13 *Lemna* spp. samples, out of 56 collected at different stations, could not be unequivocally identified by morphology and were described as ‘non-gibbous form of *L. gibba* or *L. minor’*. Plastid marker sequencing assigned all of them to *L. minor* [41]. One clone from that study (9562), originally sent by the authors to Prof. K.J. Appenroth at the University of Jena was then delivered to the Landolt Duckweed Collection, is one of the six clones recognized as hybrid in our work. It is therefore likely that such hybrids are often identified as ‘flat forms’ of *L. gibba* and are widely distributed in Italy. Hybrid discovery in Egypt (clone 7320) and Israel (7641) suggests they are adapted to the Mediterranean climate, while the clone collected in Hamburg Germany, (9425a), could be explained either as a recent invasion or the result of local hybridization occurred in that area. Mixed populations of *L. gibba* and *L. minor* have been repeatedly reported by De Lange in The Netherlands [9]. Despite their high morphological similarity to the parental species, *L. gibba* × *L. minor* must have some physiological and ecological features that confer increased fitness in competition with *L. gibba* in certain conditions. Therefore, physiological and ecological properties should be investigated, as well as the true geographical distribution of the hybrid must be recorded. To this end, we believe that TBP and/or the simpler PCR amplification with the gene-specific primer pairs described in this work provide a simple tool for fast identification of the hybrid.

In conclusion, hybridization could have been one of the driving mechanism of duckweed evolution as it has been observed in other plant families, amounting to an average of 25% of plant species that are known to hybridize naturally with at least one other species [42]. What was more unexpected is that such hybridization occurs in plants which are reported to reproduce clonally and rarely flower in nature. However, extensive interlineage hybridization has been recently reported also for the predominantly clonal, aquatic plant *Hydrilla verticillata* [43].

### 3.3. Lemna sect. Lemna

Phylogenetic relationships in duckweeds have been established initially by combinations of morphological, flavonoid, allozyme and plastid DNA sequences [4] and subsequently a larger dataset including 73 accessions, have been produced using both plastid and nuclear ribosomal [21]. Despite some node resolved incongruently using the two kind of molecular markers, a solid consensus tree was obtained by the combination of both. Major conflicts in the genus *Lemna* were actually found in the section *Lemna.*

The existence of hybrid species, particularly *L. ×japonica,* might explain conflicting results obtained with the use of plastid and nuclear loci and shed new light on the phylogenetic history of the genus. In this work, the phylogenetic tree of sect. *Lemna* was inferred from the alignment of two concatenated β-tubulin intron sequences, which deployed high interspecific variability, and further completed with the positioning of the two interspecific hybrid species. The tree topology was congruent with what reported by Tippery, 2015 [21], that place *L. minor, L. turionifera and L. trisulca* in a separate branch with respect to *L. gibba*, *L disperma* and *L. obscura*. Interbreeding between species from the two branches of the tree demonstrates absent or incomplete reproductive isolation mechanisms between species of sect. *Lemna*. If extended to other duckweed species, this suggests that other hybrids could have originated where different species occur, or have occurred in the past, in sympatry, possibly explaining incomplete species separation by molecular markers also in species of other allied genera like *Wolffia* [27].

### 3.4. Infraspecific Variation

Despite large phenotypic and physiological infraspecific variability observed in duckweeds [44], genetic diversity is barely detectable by plastid markers, while nuclear ribosomal sequences, such as ETS and ITS, are not easily amplified in all species [21]. Although genome-wide approaches such as genotyping by sequencing or whole genome comparison are now economically affordable, such techniques require managing of huge datasets. Our study revealed that infraspecific variation within β-tubulin intron sequences identifies haplotypes associated with a particular continent. This suggests that these sequences can detect greater genetic diversity than could be estimated using plastid markers. Further investigations based on β-tubulin introns may help revealing biogeographical patterns due to intraspecific variation in other species, e.g., *L.* ×*japonica*.

Conversely, *TUBB* intron sequences obtained for some hybrid clones revealed very high similarity to those of the parent species, suggesting either recent hybridization, or very low mutation rates, as it was shown for species in the allied genus *Spirodela* Schleiden [45].

Accurate identification and understanding of the genetic structure are important elements for application purposes or when managing populations of invasive species such as the native American species *L. minuta* Kunth, in Europe. Tubulin genes, combining highly conserved gene structure and exon sequences with highly variable intron regions, provided an effective toolkit for studying genetic relationships at both specific and subspecific level in duckweeds, providing new directions for further investigations by whole genome approaches.

## 4. Materials and Methods

### 4.1. Plant Material and Cultivation

Plant material, including 98 duckweed clones, mostly from the Landolt Duckweed Collection, and named with the four-digit code as defined by Landolt, is summarized in Table 1. Information about additional clones included in the cluster analyses of the present paper can be found in [20]. *Lemna* clones were aseptically cultivated in 90 mm Petri dishes on agarized Schenk and Hildebrandt (SH) medium (plant agar 8 g/L, Duchefa Biochemia, Haarlem, NL) with 2 g/L sucrose, at pH 5.1 [46]. Fronds were maintained at 25 °C under a 16-h photoperiod with light flux of 41–45 μmol m^−2^ s^−1^ provided by white florescent lights.

### 4.2. DNA Extraction and TBP Profiling

We provide here a brief description of the protocol; the whole procedure is described in details in [20]. 

One hundred milligrams of fresh duckweed fronds were disrupted in 2 ml tubes with three 3-mm stainless steel beads, using a TissueLyser II apparatus at a frequency of 30 Hz for 1 min. The standard protocol of the DNeasy Plant Mini Kit (Qiagen, Valencia, CA, USA) was adopted for the extraction of total DNA, then the quality and amount were determined by UV absorbance with the Nanodrop 2000 C (Thermo Fisher Scientific, Inc., Waltham, MA, USA), and DNA was stored at −20°C until used.

Thirty nanograms of gDNA was used as template for the TBP profiling (1st and 2nd intron). PCR primers, amplification protocols, as well all subsequent steps including Capillary Electrophoresis (CE) and the data collection were performed as reported [20]. Amplicon lengths (expressed in base pairs) of the CE-TBP profiles were used to analyze clones and the sorting of the numerical data were performed according to them. Both TBP 1st and 2nd intron markers (peaks size) were scored in a binary matrix (1/0, presence and absence respectively). The genetic similarity among genotypes was estimated according to the Jaccard’s index for binary data, using the open source software package Past v.4.07b [47]. The multivariate analysis including the cluster distribution and the principal component analysis (PCA) were also performed through the same software. The dendrogram was computed by the neighbor joining (NJ) algorithm, and bootstrap confidence values were obtained applying 1000 replications. Minor graphical editing was performed using the online tool Interactive Tree Of Life (iTOL) v.6.4 [48].

### 4.3. DNA Barcoding Analysis

The plastid markers *atp*F-*atp*H and *psb*K-*psb*I were amplified with a modified version of the primers according to [27]. PCR reactions were run in a total volume of 20 μL, with 1 Unit of Platinum Taq DNA Polymerase and 0.5 μM of each primer, using 20 ng of template gDNA. Reactions were carried out by incubation at 95 °C for 3 min followed by 30 cycles of 95 °C for 40 s, 52 °C for 50 s, 72 °C for 1 min and a final extension of 72 °C for 3 min. PCR products were checked on 2.0% agarose, 0.5× Tris Borate EDTA gels containing 1× Atlas ClearSight DNA Stain (Bioatlas, Tartu, EE, Estonia). PCR products were purified using the microCLEAN PCR/DNA Cleanup reagent, according to the manufacturer’s instructions (Labgene Scientific SA, Châtel-Saint-Denis, CH, Switzerland). The amplified products were forward and reverse sequenced (Microsynth, Balgach, Switzerland) and the obtained consensus sequence (contig) were considered for the alignment. The NCBI BLASTn analysis was performed for clone identification by best match analysis (http://www.ncbi.nlm.nih.gov (accessed on 14 November 2021)).

### 4.4. β-. Tubulin Intron Amplification and Sequencing

Gene-specific, cross-species primer pairs were designed on the sequence alignment of the chosen β-tubulin genes (*TUBB*1 and *TUBB*2) of *L. minor* clones 5500 and 8623 with the corresponding orthologs of *L. gibba* 7742, and tested for the simultaneous amplification of all the seven species belonging to *Lemna* sect. *Lemna*. More conserved regions encompassing intron borders were chosen for the primer design and degenerated nucleotides were introduced at polymorphic sites. *TUBB*2 cross-species primers (forward: I-FwTUBB2 5′-CCT CCA GGG TAT GCG ATC-3′ and reverse: I-Rv_11-23_1 5′- GGA ATC CTG CAM KTA AAT GAY G-3) targeted the first intron, whereas *TUBB*1 primers (forward: II-FwTUBB1 5′-CAC YCC AAG CTG TAA GWT CC-3′ and reverse: II-Rv-TUBB15′- GAT CGC CGA CTA YAA GAA ATC-3′) amplified the second intron.

Furthermore, species-specific primers for the selective amplification of the *L. gibba* (forward I-FwTUBB2 5′-CCT CCA GGG TAT GCG ATC-3′ and reverse I-RvTUBB2g 5′-AAC TTG GAA TCC TGC AAG CA-3′) and *L. minor* (forward 1_Fw_11-23_15′-TTC AGG GTA TGC GAT CTA TTC-3′ and reverse 1_Rv_11-23_1 5′- GGA ATC CTG CAM KTA AAT GAY G-3′) *TUBB*2 orthologous genes in interspecific hybrids were designed to target more specific regions of the same sequence alignment.

PCR was performed according to [20] after optimization of the annealing temperature; amplification products were purified using microCLEAN PCR/DNA Cleanup and directly sequenced on both strands. In the case of *TUBB*1 amplification on *L. gibba × L. minor* clones, the two amplified bands were cut from gel and purified with the MinElute Reaction Cleanup (Qiagen, Valencia, CA, USA). Forward and reverse sequences were inspected, manually edited where necessary, and combined in single consensus sequences.

### 4.5. Bioinformatic Sequence Analysis

Sequences were subjected to multiple sequence alignment using MUSCLE [49] implemented in MEGA X v.10.1.8 [50]. Alignments were manually edited and the evolutionary analysis was inferred by using the Maximum Likelihood method and General Time Reversible models [51]. Initial tree(s) for the heuristic search were obtained by applying the Neighbor-Joining method to a matrix of pairwise distances estimated using the Maximum Composite Likelihood (MCL) approach. The percentage of trees in which the associated taxa clustered together were estimated by bootstrap test (1000 replicates) and shown next to the tree branches. A discrete Gamma distribution was used to model evolutionary rate differences among sites (+G, 2 categories) and the rate variation model (+I) allowed for some sites to be evolutionarily invariable. The tree with the highest log likelihood was drawn to scale, with branch lengths measured in the number of substitutions per site.

### 4.6. Flower Induction

Fifty fronds for each of the 11 selected clones of *L. minor* (9977, 9942), *L. gibba* (9598, 8124) and *L. gibba* × *L. minor* (9425a, 6861, 7641, 9562, 9248, 7320) were aseptically cultured in Sterivent Low Containers 107 mm × 94 mm × 65 mm (Duchefa Biochemia, Haarlem, The Netherlands), in liquid SH medium supplemented with 2 g/L sucrose and 20 μM salicylic acid (SA). The trial was performed in the laboratory from mid-July to August for 40 days. Plastic boxes were positioned indoor in front of the window, under natural daylight (approx. 15 h day length) and avoiding direct sunlight, at room temperature (23–28 °C). Three replicates were set up for each clone, with an additional sample without SA as negative control. Flowers were observed from the 4th week.

## Figures and Tables

**Figure 1 plants-10-02767-f001:**
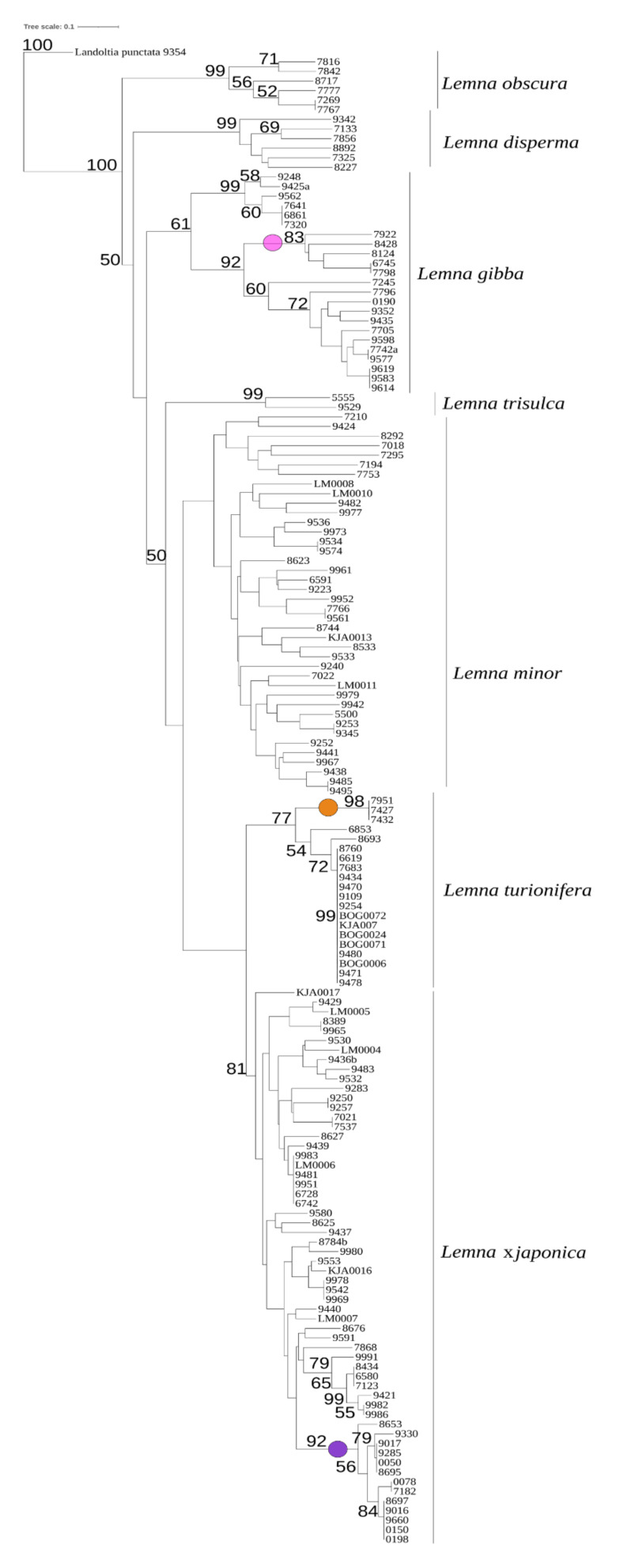
Neighbor joining similarity tree of 156 duckweed clones belonging to *Lemna* sect. *Lemna*, inferred through TBP fragment analysis (1st and 2nd intron regions). *Landoltia punctata* 9354 was used as outgroup to root the tree. The estimated bootstrap values (1000 replicates, >50%) are reported at the branch node. Colored dots highlight sub-cluster grouping clones with shared geographic origin: pink, *L. gibba* from America; orange*, L. turionifera* from East Asia; violet, *L.* ×*japonica* from East Asia).

**Figure 2 plants-10-02767-f002:**
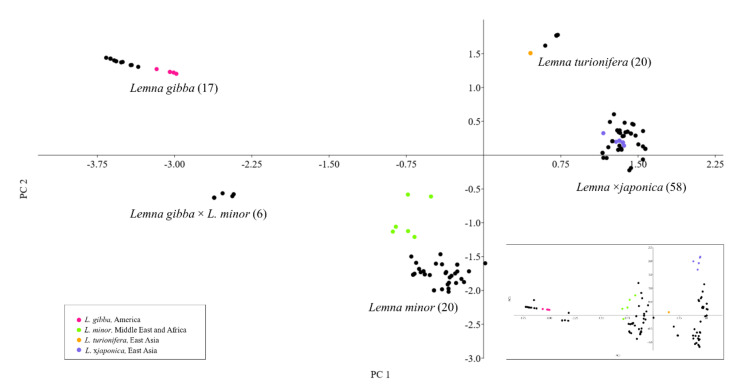
Principal components analysis (PCA) plot based on genetic distances between clones of *L. minor*, *L.* ×*japonica*, *L. gibba* and *L. turionifera*, inferred from TBP analysis (PC1 and PC2). In the insert, the plot of PC1 and PC3 shows the distribution of *L.* ×*japonica* clones. Clone IDs are omitted and the total number of analyzed clones per species is reported in brackets. Colored dots highlight only clones of each species forming the subclusters with shared geographic origin shown in Figure 1. The color code of dots is in accordance.

**Figure 3 plants-10-02767-f003:**
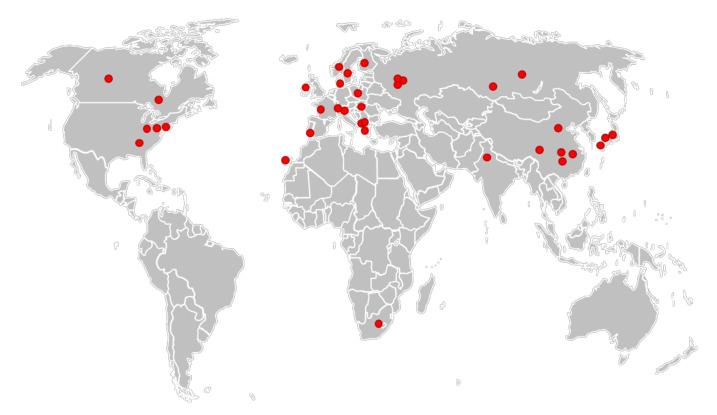
Geographical distribution of the analyzed clones of *L.* ×*japonica*.

**Figure 4 plants-10-02767-f004:**
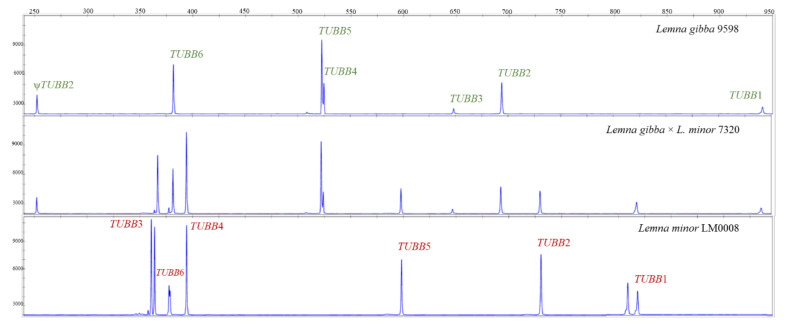
TBP profiles of representative clones of *L. gibba* × *L. minor* hybrids and the two putative parental species *L*. *minor* and *L. gibba*. Peak size is expressed in base pairs and peak height in Relative Fluorescence units. *TUBB* loci corresponding to each peak, as deduced from expected amplicon size, are indicated. Doublets indicate length variant heterozygosity at *TUBB*3 and *TUBB*1 loci in LM0008.

**Figure 5 plants-10-02767-f005:**
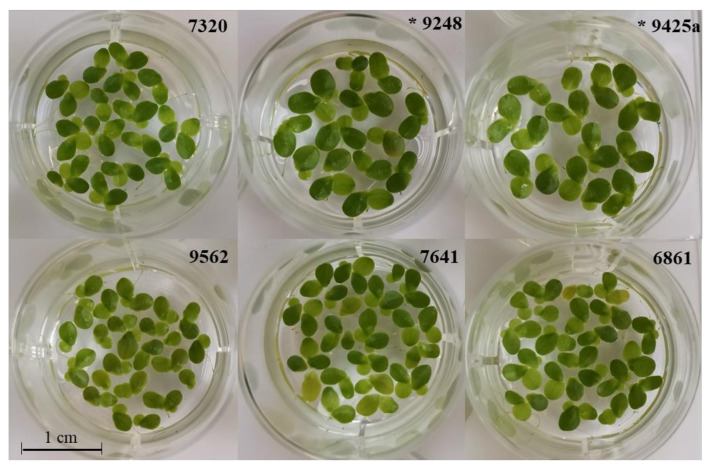
Cultures of each of the six hybrid clones *L. gibba × L. minor:* 9562, 7641, 6861, 7320, 9425a and 9248. Asterisks (*) indicate clones with *L. gibba* as the female maternal parental.

**Figure 6 plants-10-02767-f006:**
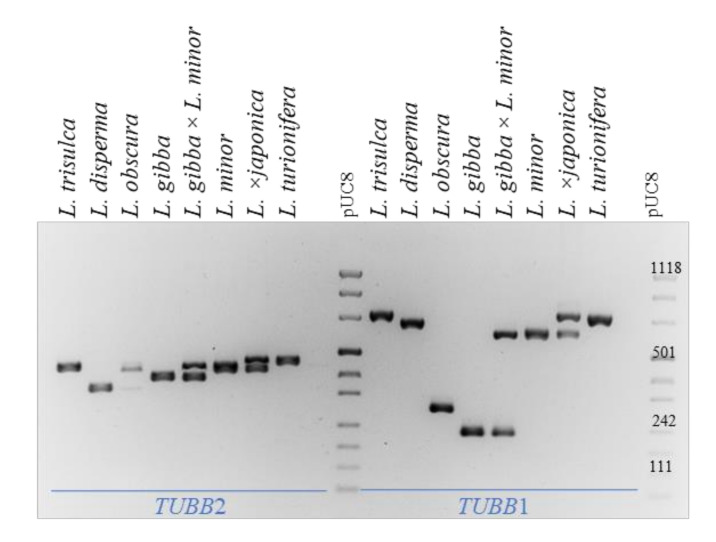
Cross-species PCR amplification of *TUBB*2 and *TUBB*1 in each species and hybrids of *Lemna* sect. *Lemna*.

**Figure 7 plants-10-02767-f007:**
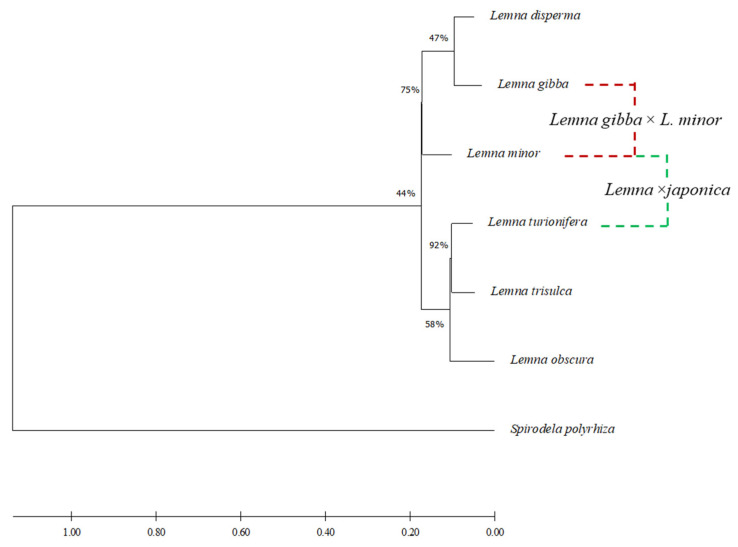
Maximum likelihood tree inferred from the alignment of concatenated loci *TUBB*2 and *TUBB*1 for all species of *Lemna* sect. *Lemna*. Bootstrap values are shown on the specific branches. Interspecific hybrids were manually inserted on the tree; relative branches are not to scale.

**Table 1 plants-10-02767-t001:** List of the plant material and reclassification of clones by TBP analysis.

Clone ID	Collection	Continent/Region	Country	Classification System	
				Morphological Characters	TBP Analysis
0050	Landolt Collection	Asia	China	*L. japonica*	*L.* ×*japonica*
0078	Landolt Collection	Asia	China	*L. japonica*	*L.* ×*japonica*
0150	Landolt Collection	Asia	China	*L. japonica*	*L.* ×*japonica*
0190	Landolt Collection	North America	USA	*L. japonica*	*L. gibba **
0198	Landolt Collection	Asia	China	*L. japonica*	*L.* ×*japonica*
6580	Landolt Collection	North America	USA	*L. minor*	*L. ×japonica **
6591	Landolt Collection	North America	USA	*L. minor*	*L. minor*
6619	Jena University	North America	USA	*L. turionifera*	*L. turionifera*
6728	Jena University	North America	USA	*L. turionifera*	*L. ×japonica **
6742	Landolt Collection	North America	USA	*L. japonica*	*L.* ×*japonica*
6745	Landolt Collection	North America	USA	*L. gibba*	*L. gibba*
6853	Jena University	North America	Canada	*L. turionifera*	*L. turionifera*
6861	Landolt Collection	Europe	Italy	*L. gibba*	*L. gibba* × *L. Minor **
7018	Landolt Collection	Asia	Turkey	*L. minor*	*L. minor*
7021	Landolt Collection	Europe	Spain	*L. gibba*	*L. ×japonica **
7123	Landolt Collection	North America	Canada	*L. minor*	*L. ×japonica **
7182	Landolt Collection	East Asia	Japan	*L. japonica*	*L.* ×*japonica*
7295	Landolt Collection	Africa	Libya	*L. minor*	*L. minor*
7320	Landolt Collection	Africa	Egypt	*L. gibba*	*L. gibba* × *L. Minor **
7427	Landolt Collection	East Asia	Japan	*L. turionifera*	*L. turionifera*
7432	Landolt Collection	East Asia	Japan	*L. japonica*	*L. turionifera **
7537	Landolt Collection	Europe	Spain	*L. gibba*	*L. ×japonica **
7641	Landolt Collection	Asia	Israel	*L. gibba*	*L. gibba* × *L. Minor **
7683	Landolt Collection	Asia	South Korea	*L. turionifera*	*L. turionifera*
7705	Landolt Collection	India	India	*L. gibba*	*L. gibba*
7767	Landolt Collection	Oceania	Australia	*L. disperma*	*L. disperma*
7777	Landolt Collection	Oceania	Australia	*L. disperma*	*L. disperma*
7798	Landolt Collection	South America	Peru	*L. gibba*	*L. gibba*
7816	Landolt Collection	Oceania	Australia	*L. disperma*	*L. disperma*
7856	Landolt Collection	North America	USA	*L. obscura*	*L. obscura*
7868	Jena University	Europe	Ireland	*L. japonica*	*L.* ×*japonica*
7922	Landolt Collection	South America	Argentina	*L. gibba*	*L. gibba*
7951	Landolt Collection	Asia	China	*L. turionifera*	*L. turionifera*
8227	Landolt Collection	North America	USA	*L. obscura*	*L. obscura*
8428	Landolt Collection	Europe	Switzerland	*L. gibba*	*L. gibba*
8434	Landolt Collection	North America	Canada	*L. minor*	*L. ×japonica **
8653	Landolt Collection	Asia	China	*L. japonica*	*L.* ×*japonica*
8697	Landolt Collection	East Asia	Japan	*L. japonica*	*L.* ×*japonica*
8717	Landolt Collection	Oceania	Australia	*L. L. disperma*	*L. disperma*
8760	Landolt Collection	Europe	Czech Republic	*L. turionifera*	*L. turionifera*
8892	Landolt Collection	North America	USA	*L. obscura*	*L. obscura*
9016	Landolt Collection	East Asia	Japan	*L. japonica*	*L.* ×*japonica*
9109	Jena University	Europe	Poland	*L. turionifera*	*L. turionifera*
9223	Landolt Collection	Europe	United Kingdom	*L. minor*	*L. minor*
9240	Landolt Collection	Europe, Asia	Russia	*L. minor*	*L. minor*
9248	Landolt Collection	Europe	Italy	*L. gibba*	*L. gibba* × *L. Minor **
9250	Landolt Collection	Europe	Finland	*L. japonica*	*L.* ×*japonica*
9253	Landolt Collection	Europe	Finland	*L. minor*	*L. minor*
9254	Landolt Collection	Europe	Finland	*L. turionifera*	*L. turionifera*
9285	Landolt Collection	Asia	China	*L. japonica*	*L.* ×*japonica*
9330	Landolt Collection	Asia	China	*L. japonica*	*L.* ×*japonica*
9345	Landolt Collection	Europe	Switzerland	*L. minor*	*L. minor*
9352	Landolt Collection	Europe	Albania	*L. gibba*	*L. gibba*
9421	Landolt Collection	North America	USA	*L. japonica*	*L.* ×*japonica*
9424	Landolt Collection	Europe	Germany	*L. minor*	*L. minor*
9429	Jena University	Europe, Asia	Russia	*L. turionifera*	*L. ×japonica **
9435	Landolt Collection	Europe	Albania	*L. gibba*	*L. gibba*
9438	Landolt Collection	Europe	Czech Republic	*L. minor*	*L. minor*
9439	Landolt Collection	Europe	Germany	*L. minor*	*L. ×japonica **
9470	Landolt Collection	Europe	United Kingdom	*L. turionifera*	*L. turionifera*
9471	Jena University	Europe	United Kingdom	*L. turionifera*	*L. turionifera*
9478	Jena University	Europe	Poland	*L. turionifera*	*L. turionifera*
9480	Landolt Collection	Europe, Asia	Russia	*L. turionifera*	*L. turionifera*
9482	Landolt Collection	Europe	Italy	*L. minor*	*L. minor*
9483	Landolt Collection	Europe	Albania	*L. minor*	*L. ×japonica **
9485	Landolt Collection	Europe	Ireland	*L. minor*	*L. minor*
9532	Landolt Collection	Europe	Macedonia	*L. minor*	*L. ×japonica **
9534	Landolt Collection	Europe	Germany	*L. minor*	*L. minor*
9542	Landolt Collection	Europe	Italy	*L. minor*	*L. ×japonica **
9561	Landolt Collection	Europe	Sweden	*L. minor*	*L. minor*
9562	Jena University	Europe	Germany	*L. gibba*	*L. gibba* × *L. Minor **
9574	Landolt Collection	Oceania	New Zealand	*L. minor*	*L. minor*
9577	Landolt Collection	Europe	Italy	*L. gibba*	*L. gibba*
9591	Landolt Collection	Europe	Hungary	*L. gibba*	*L. ×japonica **
9598	Landolt Collection	Europe	Germany	*L. gibba*	*L. gibba*
9660	Landolt Collection	Asia	China	*L. japonica*	*L.* ×*japonica*
9942	Landolt Collection	Europe	Norway	*L. minor*	*L. minor*
9951	Landolt Collection	Europe	France	*L. gibba*	*L. ×japonica **
9952	Landolt Collection	Europe	France	*L. minor*	*L. minor*
9961	Landolt Collection	Europe	Germany	*L. minor*	*L. minor*
9965	Landolt Collection	Europe	Switzerland	*L. gibba*	*L. ×japonica **
9967	Landolt Collection	Europe	Switzerland	*L. minor*	*L. minor*
9969	Landolt Collection	Europe	Switzerland	*L. minor*	*L. ×japonica **
9973	Landolt Collection	Europe	Germany	*L. minor*	*L. minor*
9977	Landolt Collection	Europe	Germany	*L. minor*	*L. minor*
9978	Landolt Collection	Europe	Switzerland	*L. minor*	*L. ×japonica **
9979	Landolt Collection	Europe	Germany	*L. minor*	*L. minor*
9980	Landolt Collection	Europe	Germany	*L. minor*	*L. ×japonica **
9982	Landolt Collection	North America	USA	*L. japonica*	*L.* ×*japonica*
9983	Landolt Collection	Europe	Switzerland	*L. japonica*	*L.* ×*japonica*
9986	Landolt Collection	North America	USA	*L. minor*	*L. ×japonica **
9991	Landolt Collection	North America	USA	*L. japonica*	*L.* ×*japonica*
8784b	Landolt Collection	Europe	Sweden	*L. japonica*	*L.* ×*japonica*
9425a	Landolt Collection	Europe	Italy	*L. gibba*	*L. gibba* × *L. Minor **
BOG0024	Greifswald University	Europe	Germany	*L. turionifera*	*L. turionifera*
BOG0071	Greifswald University	Europe	Germany	*L. turionifera*	*L. turionifera*
BOG0072	Greifswald University	Europe	Germany	*L. turionifera*	*L. turionifera*
KJA007	Jena University	Europe, Asia	Russia	*L. turionifera*	*L. turionifera*

* reclassified according to TBP analysis.

**Table 2 plants-10-02767-t002:** Misidentification rate: correlation between morphological classification and TBP analysis.

		TBP						
	Species	*L. minor*	*L. turionifera*	*L.* ×*japonica*	*L. gibba*	*L. gibba* × *L. minor*	Tot. n	% Incorrect
**morphology**	** *L. minor* **	21		11			32	34.3
	** *L. turionifera* **		16	2			18	11.1
	** *L.* ** **×*japonica***		1	19	1		21	9.5
	** *L. gibba* **			5	9	6	20	55.0
	** *L. gibba* ** **× *L. minor***					0	0	0.0
	**tot. n**	21	17	37	10	6	91	28.6
	**% incorrect**	0.0	5.9	48.6	10.0	100.0	28.6	

**Table 3 plants-10-02767-t003:** *TUBB*2 polymorphic sites and corresponding haplotypes.

	Haplotype	Polymorphic Sites														Alignment Length (bp)
		35	45	86	175	177	184	197	206	249	286	316	330	337	366	407
** *L. minor* **	M1	T	A	T	G	C	C	T	T	A	C	G	G	G	A	
	M2	C	G	G	T	G	T	A	G	T	T	A	A	A	T	
	M3	T	A	T	T	C	C	T	T	A	C	G	G	G	A	
	M4	T	A	T	T	C	C	T	T	A	C	A	G	G	A	
		40	68	109	224	250	255	302	329							380
** *L. gibba* **	G1	T	T	C	G	C	T	A	G							
	G2	--	T	C	G	T	T	A	G							
	G3	--	A	T	--	T	C	C	T							

Position numbers are given from the beginning of the aligned region.

**Table 4 plants-10-02767-t004:** Association of the *TUBB* haplotypes to the analyzed clones.

	Clone	Origin	Haplotype	SNPs		Clone	Origin	Haplotype	SNPs
** *L. minor* **	5500	Ireland	M1	0	** *L. gibba* **	7742a	Italy	G1	0
	9961	Germany	M1	0		9598	Italy	G1	0
	9424	Germany	M1	0		6745	California—USA	G3	6
	7194	Uganda	M2	12		8124	Arizona—USA	G3	6
	7753	Ethiopia	M2	12		7705	India	G2	2
	7210	S. Africa	M2	12		190	USA	G2	2
	9495	Norway	M3	1		7796	Italy	G2	2
	LM0011	Russia	M3	1		9583	Poland	n.d	--
	7766	New Zealand	M3	1		7922	Argentina	n.d	--
	9536	Germany	M4	2		9614	Poland	n.d	--
	7022	Spain	M4	2		9619	Albany	n.d	--
	9482	Italy	M4	2		7245	S. Africa	n.d	--
	9942	Norway	M4	2		8428	Switzerland	n.d	--
	9533	Macedonia	M4	2	** *L. gibba* ** **× *L. minor***	7320	Egypt	G2 M4	
	8744	Albania	M4	2		6861	Italy	G2 M4	
	LM0010	Italy	M4	2		9562	Italy	G2 M4	
	LM0008	Russia	M4	2		9425a	Germany	G2 M4	
	9252	Finland	M4	2		7641	Israel	G2 M4	
	8292	Iran	M4	2		9248	Italy	G2 M1	

The number of substitutions is reported with respect to *L. minor* 5500 and *L. gibba* 7742a taken as references. n.d, sequence with overlapping peaks in the chromatogram.

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
