# Peer review of "New Insights into Interspecific Hybridization in *Lemna* L. Sect. *Lemna* (Lemnaceae Martinov)"

_plants, 2021, doi:10.3390/plants10122767_

Round 1
Reviewer 1 Report
General comments:
The authors have provided excellent evidence to prove the misidentification of some lines of L. gibba while they are interspecific hybridization between L. minor and L. gibba.
Detailed comments and suggestions:
- This is a rather long but excellent manuscript, the experiments were well designed, the data was highly reliable.
- There are several typos
Page 3 – line 133: with instead of whit
Page 11 – line 279: maternal instead of female parental
Page 11 – line 284: a synonym of (remove “with”)
Tab. 4: it must read Heterozygous (not Heterozigous)
Page 16 – line 477: and instead of nd
Page 18 – line 572: Electrophoresis instead of Elctrophores
Conclusion:
The manuscript could be accepted after minor revision according to the recommendations of reviewers and editors.

Author Response
We thank the reviewer for his appreciation of our work and introduced all the requested corrections
Reviewer 2 Report
The submission by Braglia and coauthors, entitled “New insights into interspecific hybridization in Lemna L. sect. Lemna (Lemnaceae Martinov)”, provides an interesting perspective about potential hybridization among duckweed species. The data come from length variants in the β-tubulin gene, and the patterns are consistent with what would be expected after hybridization. I would say that the evidence is sufficient to support the authors’ claim of hybridization, although more convincing evidence would come from sequence polymorphisms (not just length polymorphisms) that could be identified individually by subcloning (such as has been done to confirm hybrids in many other plant groups. Nonetheless, the prospect of hybrid duckweeds is interesting and worth reporting.
Below I offer some specific comments and suggestions:
In the list of authors, only one is provided with an address. If the other authors are at the same address, they should receive the same numeric identifier.
The method of “Tubulin Based Polymorphism (TBP)” has earlier origins before its use in duckweeds. A paper such as Bardini et al. 2004 (https://doi.org/10.1139/g03-132) should be cited as evidence of the method’s broader applicability.
Line 16: I think the term “fuzzy” could be replaced with a more precise description of species boundaries.
I recommend that the taxonomic disposition of Lemna japonica should be treated consistently throughout the manuscript. The species is introduced in the abstract as being difficult to distinguish from L. minor. However, the authors clearly favor the idea that L. japonica is a hybrid species with L. minor and L. turionifera as its parent species (following Braglia et al. 2021: https://doi.org/10.3389/fpls.2021.625670). I think the manuscript could introduce L. japonica as a hybrid species (at least putatively based on previous TBP evidence) and set this up as a precedent for the results they are presenting in the current manuscript.
Line 29: Wolffia and Wolffiella need to be italicized. Also, “Woffia” is a typographical error. Also, the taxonomic authority for Wolffia is provided much later in the manuscript, at line 525, when this is the first mention of the name.
Line 34: This sentence is a fragment. The phrase “among which…” is never resolved with another verb.
Line 38: I am not familiar with the concept of a “hard genus”. I recommend the authors use a different descriptor here. The authors seem to intend the term “hard” to mean “difficult”, but the term also could be interpreted to mean “firm” (a double meaning of “hard” in English), and therefore it is not a useful term for several reasons.
Line 44: One might think that “flat” refers to dried herbarium material, but it is not perfectly clear. Please use precise language to avoid misunderstandings. Line 48 mentions “permanently flat forms”, and this leaves me puzzled. Please explain what you mean by “flat” in this context. (See also line 269.)
Throughout the manuscript, the genus name “Lemna” is spelled out rather than abbreviated, when the abbreviated form “L.” would suffice.
Line 71: The citation [18] also includes the hypothesis about L. japonica being of hybrid origin, so I recommend placing the citation at the end of the sentence.
Line 77: Spell out the words that make up the abbreviation “TBP”. For example: “based on intron length polymorphism in the β-tubulin genes (also known as tubulin-based polymorphism, or TBP)”.
Line 115: It might be helpful to provide a website or other information about the LDC, for example to mention that it is based in Zürich, Switzerland.
Line 133: Typo: “whit”
Line 134: It might be useful to spell out the “capillary electrophoresis” part of “CE-TBP” here.
Table 1 lists Europe, Asia, and Eurasia as continents/regions, as though they are mutually exclusive. I suggest using “Eurasia” for all of these.
There is a typographical error in Table 1 for clone 8434: “Nort America”
Figure 1 is rather pixelated and may need to be replaced with a higher-resolution image. Also, the text appears to be stretched vertically, and this could be corrected by stretching the image in width (and thus filling more of the available space on the page.
The colored dots on Fig. 1 are not particularly helpful, because the reader needs to discover on their own which geographic regions are indicated by which color. A textual label for geography might be more helpful. (A color legend is provided for Fig. 2 but not for Fig. 1. For some reason, there are two colors that both correspond to “East Asia” in the Fig. 2 legend, and many dots that have no color.)
The taxon labels on Fig. 1 do not suggest any hybridization except for L. japonica. Therefore, I question the utility of showing such a result when the manuscript largely focuses on hybridization. Line 155 indicates that the bootstrap support is (understandably) low for hybrids because they share features in common with both parental species, and this might be an argument for abandoning the tree as a useful depiction of results.
Line 154: Please be more precise when you say that clones in the L. minor cluster have “similar pattern”. Similar to one another, or similar to the L. japonica pattern?
Line 171: There is a paragraph that appears to be left over from the document template: “This section may be divided by subheadings. It should provide a concise and precise description of the experimental results, their interpretation, as well as the experimental conclusions that can be drawn.”
Line 218: I don’t understand what the authors mean by “numerical consistency”.
Line 239: You cannot arbitrarily assign these names, because they are identical to regulated gene names. From Fig. 4 it appears that the arbitrary names occur in decreasing order of intron length, but I believe that the arbitrary peak for “TUBB3” is not the same gene as the homolog of TUBB3 in other plants (e.g., https://www.uniprot.org/uniprot/Q9ZRB0). Please choose a different way to refer to the length variants.
The Fig. 4 data show an additive combination of genetic elements from the putative parents, a pattern you would expect to result from allopolyploidy. The genetic elements are not allelic, where only some of them would be passed on to the hybrid. I think this pattern might be worthy of mention, possibly in light of any known chromosome counts.
Line 282: What do you mean to say that L. symmeter was “(invalidly) described”? This name is not mentioned in the International Plant Names Index, so perhaps it was not validly published. (I cannot retrieve the original publication to investigate.)
Table 4: The spelling should be “Heterozygous”. Also, can you provide more detail about the “heterozygous” haplotypes? How is “heterozygous” different from the cases where two allele names are provided (e.g., G2 M4)? Also, “haplotype” may not be the appropriate term to use here, if there are multiple alleles involved (the “haplo” in “haplotype” refers to a haploid ploidy level).
Line 355: The abbreviation “SNP” is commonly used, but it should still be spelled out once as “single nucleotide polymorphism”.
Line 363: You mean to use “haplotype” instead of “holotype”.
Line 431: The proper spelling for the genus is “Senecio”.
Line 455: It may be premature to assign “high probability” to this speculation about the identity of “L. symmeter”.
Line 572: Typographical error: “Elctrophoresis”
Line 602: Apparently “hortologs” should be “orthologs”. (see also Line 614: “hortologous”)
